# Water Vapor Content Retrieval Under Cloudy Sky Conditions from SWIR Satellite Measurements in the Context of C<sup>3</sup>IEL Space Mission Project

Raphaël Peroni<sup>1,+</sup>, Guillaume Penide<sup>1</sup>, Céline Cornet<sup>1</sup>, Olivier Pujol<sup>1</sup>, and Clémence Pierangelo<sup>2</sup>

Correspondence: Céline Cornet (celine.cornet@univ-lille.fr)

**Abstract.** A retrieval algorithm of integrated water vapor content above cloud, using shortwave infrared observations, is developed and evaluated through idealized and realistic atmospheric profiles, with its application currently limited to oceanic regions and latitudes within  $\pm 60^{\circ}$ . Water vapor plays a crucial role in cloud formation and development, particularly in the formation of clouds resulting from convective processes. The resulting convective cloud locally influences the spatio-temporal variability of atmospheric water vapor content, through exchanges between cloud and its immediate environment. Therefore, a better understanding of the water vapor content above and around clouds is necessary to improve our comprehension of interactions between water vapor and cloud to better constrain Large-Eddy Simulations (LES) and numerical weather forecasting models. The algorithm was developed for the Cluster for Cloud evolution, ClImatE and Lightning, C<sup>3</sup>IEL space mission project. This mission, scheduled for 2028, aims to enhance our knowledge of the 3D convective cloud development velocities, the electrical activity associated with convective systems, and the water vapor content above and around the cloud. The retrieval algorithm presented in this study uses a Bayesian probabilistic approach, the optimal estimation method. The atmosphere is assumed to be composed of homogeneous plane-parallel layers, and synthetic radiance datasets were generated to test the developed retrieval algorithm. The feasibility of retrieving the integrated water vapor content above the cloud over the ocean from SWIR radiances is shown to have, under idealized vertically homogeneous cloud profiles, absolute errors less than 2 kg.m<sup>-2</sup> for optically thick clouds or when the integrated water vapor content is below 20 kg.m<sup>-2</sup> and less than 1 kg.m<sup>-2</sup> for very thick clouds with an optical thickness exceeding 150. Tests using realistic water vapor and cloud extinction profiles that present non-homogeneous vertical distributions show that integrated water vapor content above water type clouds could be retrieved with a Root-Mean-Square Error (RMSE) related to cloud vertical penetration of approximately less than 1 kg.m<sup>-2</sup> except for optically thin and low-level clouds (cloud optical thickness less than 50 and cloud top height less than 2 km). For very low water vapor content encountered in the presence of high deep convective clouds, the retrieval algorithm tends to systematically overestimate the retrieved water vapor content due to an overestimation of the cloud extinction profile in the upper part of the cloud in the inversion model.

<sup>&</sup>lt;sup>1</sup>Univ. Lille, CNRS, UMR 8518 - LOA - Laboratoire d'Optique Atmosphérique, F-59000 Lille, France

<sup>&</sup>lt;sup>2</sup>Centre National d'Etudes Spatiales (CNES), 18 avenue Edouard Belin, 31401 TOULOUSE Cedex 9, France

<sup>&</sup>lt;sup>+</sup>Now at Department of Earth and Atmospheric Sciences, Université du Québec à Montréal, Montreal, Canada

### 1 Introduction

Clouds play a significant role in Earth's energy balance, as they can induce a positive or negative radiative forcing whose uncertainty is still important (*e.g.*, Stocker et al., 2013; Masson-Delmotte et al., 2021). Depending on their latitude, characteristics, altitude and temperature, their radiative forcing can be different (*e.g.*, Ramanathan et al., 1989; Harrison et al., 1990). In fact, lower tropospheric clouds, which are composed primarily of liquid water, tend to reflect solar radiation at all wavelengths, resulting in a cooling effect (*e.g.*, Fermepin and Bony, 2014), named the parasol effect (*e.g.*, Crutzen and Ramanathan, 2003). In contrast, solar radiation goes through upper thin tropospheric clouds, such as *cirrus* but these thin high clouds absorb thermal radiation from the Earth and re-emit infrared radiation towards space at a lower temperature due to their cold temperature, thereby enhancing the greenhouse effect (*e.g.*, Jensen et al., 1996; McFarquhar et al., 2000; Lee et al., 2009; Schmidt et al., 2010). A more detailed description of *cirrus* type clouds and their effects on the Earth-Atmosphere system can be seen in Lynch (2002). Deep convective clouds combines both effect as they reflect a significant amount of incoming solar radiation back into space and emit infrared radiation at a low temperature due to the high level of their top. It is therefore essential to better understand cloud development in order to accurately assess their radiative impact on the Earth's energy balance. As an example, Bony et al. (2015) address the scientific community with several questions aimed at emphasizing the importance of a better understanding of the role of cloud feedbacks and convective organization on climate, as well as the factors that influence cloud formation.

Cloud formation and development depend on the amount of water vapor available in the atmosphere. Indeed, at a given temperature, saturation is reached when the water vapor partial pressure equals the saturation vapor pressure. Water vapor will then condense either on Cloud Condensation Nuclei (CCN) to form new cloud droplets or on Ice Nuclei (IN), to form new ice crystals. The latent heat released during water vapor condensation and cloud formation not only disturbs the thermal structure of the atmosphere (*e.g.*, Trenberth and Smith, 2005; Schneider et al., 2010) but also fuels cloud development through a chain reaction.

In the free troposphere, humidity influences the dynamical development of clouds through entrainment and detrainment processes. Convection processes, in turn, contribute significantly to the redistribution of energy and water vapor in the atmosphere (*e.g.*, Blyth, 1993). Humidity above and around clouds is therefore an essential parameter in the process of cloud development, particularly in the case of convective clouds.

Many spaceborne remote sensing instruments have been developed to retrieve water vapor across various spectral domains. Microwave sounders such as the Sounder for Probing Vertical Profiles of Humidity (SAPHIR) aboard the French-Indian satellite MEGHA-TROPIQUES (*e.g.*, Desbois et al., 2007), or AMSU (the Advanced Microwave Sounding Unit) aboard the NOAA satellite, make it possible to conduct measurements at all weather conditions and provide either humidity profiles or information on total water vapor content at a spatial resolution of 12 km at nadir for SAPHIR (*e.g.*, Rao et al., 2013), 48 km at nadir for AMSU-A, and 16 km for AMSU-B (*e.g.*, Rosenkranz, 2001; Karbou et al., 2005). Humidity profile retrieval in clear sky conditions or above thick clouds are also performed by the Infrared Atmospheric Sounding Interferometer (IASI) instrument operating in the thermal infrared (TIR) with a spatial resolution of 8 km (*e.g.*, Schlüssel and Goldberg, 2002; Hilton et al.,

2012). These instruments give a vertical information on the water vapor content in the atmosphere but their limitation to study cloud and water vapor interactions lies in their low spatial resolutions and not contiguous pixels, which do not allow for precise examination of these interactions.

60 Near-Infrared (NIR) or Shortwave Infrared (SWIR) imagers allow to derive water vapor content at a better spatial resolution under clear sky conditions through the differential absorption method, either with airborne measurements (e.g., Bouffiès et al., 1997), or from spaceborne sensors such as the POLarization and Directionaly of Earth Reflectance instrument (POLDER: e.g., Vesperini et al., 1999), the Medium Resolution Imaging Spectrometer (MERIS: e.g., Bennartz and Fischer, 2001), the MODerate resolution Imaging Spectroradiometer (MODIS: e.g., Gao and Kaufman, 2003) or the Ocean and Land Colour Instrument (OLCI: Preusker et al., 2021). All of these retrieval algorithms propose parameterizations that link the Total Column Water Vapor (TCWV), in kg.m<sup>-2</sup>, to the ratio of NIR or SWIR spectral bands within absorbing and non-absorbing channels. However, in cloudy sky conditions, these parameterizations are no longer sufficient as clouds interact with the measurements. Indeed, in this particular spectral domain, clouds are not transparent to radiation (as it is in the microwave domain). Moreover, as clouds do not act as a perfect reflector, radiation penetrates the cloud and gets scattered, effectively extending the radiation 70 path through the atmosphere and consequently increasing absorption by water vapor or any other absorbing gas. Albert et al. (2001) demonstrates the feasibility of retrieving water vapor content above cloud in the SWIR domain, over various types of surface, and in the presence of low and optically thick clouds, applying the differential absorption method to simulations conducted in the context of the POLDER and MERIS instruments. They demonstrated with simulations that the absorption of solar radiation within a water vapor absorption band is influenced by the "radiation path", modified by the presence of clouds. On one hand, multiple scattering increases the path length traveled by radiation, leading to higher absorption by water vapor and consequently, a higher retrieved water vapor content. On the other hand, the presence of clouds reduces or stops the influence of lower atmospheric layers, resulting in reduced overall absorption (Albert et al., 2001). They conclude that retrieving water vapor above cloud is feasible for high values of cloud optical thickness. By adjusting the path in their method and applying it to POLDER measurements, they report an average root mean square error (RMSE) of 1.8 kg.m<sup>-2</sup> over ocean, as compared with radio-sounding data. (Vidot et al., 2009) conducted a feasibility study using an optimal estimation method within the framework of the Orbiting Carbon Observatory (OCO) sensor, quantifying various type of errors to demonstrate the potential for retrieving column-averaged carbon dioxide mixing ratios over liquid water clouds above the ocean. Similarly, (Schepers et al., 2016) used SWIR measurements from the Greenhouse Gas Observing Satellite (GOSAT) to simultaneously retrieve the total columns of methane and carbon dioxide above clouds, along with cloud properties.

In this study, the objective is to retrieve the integrated water vapor content above cloud using SWIR satellite observations with high spatial and temporal resolution over the ocean in the context of the Cluster for Cloud evolution, ClImatE and Lightning (C<sup>3</sup>IEL) mission. The main advantage that can be exploited is the knowledge of the cloud top height retrieved with good accuracy thanks to the CLOUD radiometers (Dandini et al., 2022). Firstly, in section 2, we provide an overview of the study's context, specifically the space mission project C<sup>3</sup>IEL that contextualizes our work. Then in section 3, we discuss the method employed in the developed retrieval algorithm. Section 4 shows the sensitivity of the C<sup>3</sup>IEL water vapor channels to

the integrated water vapor content above cloud ( $IWV_{AC}$ ). In section 5, the algorithm is tested first under idealized cloudy atmospheric conditions, then under realistic conditions. Finally, section 6 summarizes the main findings and perspectives.

# 2 The C<sup>3</sup>IEL space mission project

The space mission project named as the Cluster for Cloud evolution, ClImatE and Lightning, C<sup>3</sup>IEL (Rosenfeld et al., 2022) started in 2016 through a partnership between the French space agency (CNES) and the Israeli Space Agency (ISA). The mission is under development and is scheduled to be launched at the end of 2028, with a minimum expected duration of two years that could be extended to three years. Its primary objective is to explore the dynamical development of convective clouds, including *cumulus congestus* and *cumulonimbus* clouds. This will be achieved by gathering data at high spatial and temporal resolutions. Indeed, the satellites will rotate to track the same scene for 200 seconds and capture a sequence of 11 acquisitions of two simultaneous observations, C<sup>3</sup>IEL consist of a pair of satellites operating in tandem, distanced by about 150 km following a sun-synchronous orbit at about 1:30 PM local time at the equator. The altitude of the orbit is expected to be between 600 and 700 km. The underlying measurement principle is represented in figure 1. The viewing angles for each satellite will be approximately  $\pm 50^{\circ}$ ,  $-42^{\circ}$ ,  $-32^{\circ}$ ,  $-20^{\circ}$ , and  $-7^{\circ}$  on each side of the observed scene. Additionally, the first satellite will include a  $-55^{\circ}$  angle, while the second satellite will include a  $+55^{\circ}$  angle. Between each sequence of acquisitions, the satellites will have to return to their initial attitude, which implies a limited number of 4 sequences per orbit. The latitudes of these observation sequences will be chosen according to climatology of convective clouds. This strategy of observations will provide (1) the 3D envelope of convective clouds and their vertical/horizontal development velocities (Dandini et al., 2022) using the visible imagers named CLOUD, measuring at 670 nm with a high spatial resolution (20 m at nadir), (2) the associated electrical activity generated by convective processes with the instruments Lightning Optical Imager and Photometers (LOIP) consisting in visible imagers measuring at 777 nm with a spatial resolution of 140 m at nadir and three photometers at 337, 391 and 777 nm and (3) the water vapor content above and around convective clouds using the three shortwave infrared (SWIR) imagers, with a spatial resolution of 125 m at nadir.

This paper focuses on the development of a retrieval algorithm for the integrated water vapor content above cloud  $(IWV_{AC})$  from three SWIR water vapor imagers on each satellite. Figure 2 shows the water vapor transmission spectrum in this spectral range and the three spectral bands selected for the  $C^3$ IEL water vapor imagers. The first band (1) is a non-absorbing band centered at  $1.04~\mu m$ ; the second band (2) is a moderately absorbing band centered at  $1.13~\mu m$ ; the third one is a highly absorbing band centered at  $1.37~\mu m$ . These three bands have a spectral width of about  $0.04~\mu m$ .

### 3 Methodology

100

105

120

This section introduces the retrieval scheme used in our study. It follows an Optimal Estimation Method (OEM) scheme (Rodgers, 2000), with the aim to get the 1D equivalent cloud optical thickness (COT) and the integrated water vapor above cloud ( $IWV_{AC}$ ). The OEM is a Bayesian statistical approach commonly used in remote sensing in order to estimate at-

Figure 1. Illustration of the principle of the Cluster for Cloud evolution, ClImatE and Lightning (C<sup>3</sup>IEL) observations.

**Figure 2.** Water vapor transmission spectrum in clear sky condition at nadir (red curve) and the three SWIR spectral bands for the study of water vapor in the context of the C<sup>3</sup>IEL space mission (blue rectangles). Note that rectangular spectral response functions are used, as the actual SRF are not known.

mospheric and surface properties from measurements, such as satellite remote sensing (*e.g.*, Sourdeval et al., 2013, 2015; Leonarski et al., 2020; Matar et al., 2023). The objective of the OEM is to minimize the difference between the measured and the simulated radiances, under the constraint of *a priori* knowledge about the atmosphere. Equation (1) formalizes how the

125 OEM works:

130

150

$$y = F(x, b) + \epsilon \tag{1}$$

where the vector y contains the measured radiances and F denotes the forward model including the model assumed for the retrieval and the radiative transfer code to get radiances from atmospheric properties. The state vector (x) contains the parameters to retrieve (see section 3.1), whereas b specifies the fixed parameters within the forward model (see section 3.2). Lastly, the vector  $\epsilon$  contains the errors assumed to be randomly distributed, encompassing measurement uncertainties, errors in the fixed parameters and related to the forward model. Because forward model errors are extremely hard to estimate properly, as usually done only uncertainties associated with measurements and fixed parameters are considered in this paper.

#### 3.1 Measurement and state vectors: y and x

Water vapor retrieval in the context of the  $C^3$ IEL space mission is based on the exploitation of three spectral bands in the SWIR. The non-absorbing band (centered at  $1.04 \,\mu m$ ) is sensitive to the 1D equivalent cloud optical thickness (COT), representing the medium's reflectivity, while the two other bands (centered at 1.13 and  $1.37 \,\mu m$ ), which are also sensitive to the 1D equivalent COT, are used to retrieve the  $IWV_{AC}$ .

The measurement vector contains thus the radiance values measured in the three spectral bands described above:

$$y = \begin{bmatrix} R_{1.04} \\ R_{1.13} \\ R_{1.37} \end{bmatrix} \tag{2}$$

As the C<sup>3</sup>IEL mission is still not in orbit, these radiances are simulated to develop and evaluate the algorithm using atmospheric and cloud profiles described in section 5. Measurement vector data are associated with an uncertainty of 5%. This value corresponds to the requirement on the accuracy of the radiance measured by the instrument (random error at 1 sigma), corresponding to random noise.

The state vector contains the desired parameters, the 1D equivalent COT and  $IWV_{AC}$ :

$$145 \quad x = \begin{bmatrix} 1D \text{ COT} \\ IWV_{AC} \end{bmatrix} \tag{3}$$

The uncertainty on the *a priori* knowledge is arbitrarily set to 10,000%, in order to minimize its influence and give more weight to the measurements during the retrieval process.

At the end of the OEM process, the variance-covariance matrix of the retrieved state vector is computed and gives the uncertainties on the retrieved parameters (a posteriori uncertainties, noted  $\sigma_x$ ) with the following relationship (Rodgers, 2000):

$$\sigma_x = \sqrt{\left(S_a^{-1} + K_i^T S_\epsilon^{-1} K_i\right)^{-1}} \tag{4}$$

which is the square root of the *a posteriori* variance-covariance matrix. In this expression,  $S_a$  is the *a priori* variance-covariance matrix, K the jacobian, and  $S_{\epsilon}$  the error variance-covariance matrix associated with the measurements and fixed parameters (see Appendix A for more details).

### 155 3.2 Fixed parameters: b

160

165

The cloud model defined in the developed retrieval algorithm assumes a 1D horizontally and vertically homogeneous planeparallel layer between the cloud base height (noted  $Z_b$ ) and its top altitude (noted  $Z_t$ ), horizontally infinite over the ocean. Table 1 describes the fixed parameters used in the developed retrieval algorithm and their uncertainties.

**Table 1.** Description of the fixed parameters and their uncertainties.

| Fixed parameter (b)            | Value                   | Uncertainty $(\epsilon_b)$ |
|--------------------------------|-------------------------|----------------------------|
| Surface albedo                 | 0.060                   | $\pm 0.006$                |
| cloud base height $(Z_b)$      | 0.5 or 0.71 km          | $\pm~0.32~\rm km$          |
| cloud top height $(Z_t)$       | provided by CLOUD/C3IEL | $\pm~0.04~\rm km$          |
| Droplet [Ice] effective radius | 10 [45] μm              | $\pm~5~\mu m$              |

In this study, we consistently assign a surface albedo of 0.060, representing an ocean surface, with an uncertainty chosen to be 10%. The cloud base height is set to 0.50 km for the idealized cases (section 5.2.1) or 0.71 km for the realistic cases (section 5.2) with an uncertainty of 0.32 km. These last two values represent respectively the average and standard deviation values derived from the ECMWF-IFS selected profiles for this study (section 5.2.1). The cloud top height varies based on the atmospheric profile used to simulate the measurements. In practice, it will be determined by combining data from the pair of visible imagers designed for studying the 3D envelope and development velocities. The cloud top height uncertainty is set to 0.04 km (Dandini et al., 2022). For the idealized cases, the phase of the clouds is only liquid whereas for the realistic cases, for clouds with a top altitude above 4 km, we assume they consist of two distinct phases: a liquid phase between the cloud base height ( $Z_b$ ) and Z=4 km, and an ice phase between Z=4 km and the cloud top height ( $Z_t$ ). This fixed altitude of 4 km represents the average height of the 0 °C isotherm, according to the ECMWF profiles selected for this study. The effective radius of cloud droplets is fixed at an average value of 10 µm (e.g., King et al., 2004), with an accuracy of 50%. The effective radius of ice crystals is set to an average value of 45 µm with an associated uncertainty of 5 µm. These values represent the average and standard deviation of the ice effective radius calculated using the Wyser parameterization (Wyser, 1998) applied on the whole database described in section 5.2.1.

# 3.3 Radiative transfer model: F

Our study combined an Optimal Estimation Method with the radiative transfer code ARTDECO (Atmospheric Radiative Transfer Database for Earth and Climate Observation) in order to solve the Radiative Transfer Equation (RTE) by means of the adding-doubling model (de Haan et al., 1987). ARTDECO (https://www.icare.univ-lille.fr/artdeco/, Dubuisson et al., 2016) is a tool that gathers various models and data used to simulate Earth total and polarized atmosphere radiances and radiative fluxes, from the UV to the thermal IR range ( $200~\mathrm{nm}$  to  $50~\mu\mathrm{m}$ ). It uses the homogeneous plane-parallel approximation and allows to compute aerosols and clouds optical properties.

# 180 3.4 $IWV_{AC}$ retrieval: principle and assumptions

Given the limited information on the atmospheric profile from the measurements, assumptions are made to constrain the model: molecular Rayleigh scattering is disregarded due to its negligible effect in the SWIR, and relative humidity (RH) is assumed to be 100% within the cloud. For low- and mid-level liquid clouds (Section 5.2.2), the assumption is applied with respect to liquid water. For high-level mixed-phase clouds (Section 5.2.3), relative humidity is defined as 100% with respect to liquid water below 4 km and with respect to ice above.

As explained above, the main objective of the developed retrieval algorithm is to minimize the difference between the measured radiances and the radiances simulated by the forward model. The process begins with a first guess value for the state vector including the 1D equivalent cloud optical thickness (COT) and  $IWV_{AC}$ . For the COT, the first guess value is 10 and for the  $IWV_{AC}$ , the first guess value is calculated by integrating the SAS (Sub-Arctic Summer) water vapor profile from the cloud top altitude up to 20 km. In each iteration, the state vector is adjusted to achieve simulated radiances closer to the measured one. The COT, being an input parameter in the radiative transfer code, is subject to a direct adjustment at each iteration. The  $IWV_{AC}$  follows a different approach as it represents the vertical integration of the water vapor profile above clouds used in the forward model. The adjustment consists in applying, at each iteration, a multiplicative factor  $\beta$  to the water vapor profile, noted WV(z), from the cloud top height ( $Z_t$ ) to the top of atmosphere. The resulting adjustment coefficient,  $\beta$ , is derived by calculating the ratio between the estimated  $IWV_{AC}$  at iteration i+1 and the one estimated at iteration i:

$$WV_{i+1}(z) = \beta \cdot WV_i(z) \tag{5}$$

with,

205

185

190

$$\beta = \frac{IWVAC_{i+1}}{IWVAC_i} \tag{6}$$

# 4 Sensitivity of the three spectral bands to COT and $IWV_{AC}$

In this section, we examine how the radiances simulated in the non-absorbing band (1.04  $\mu$ m) and in the water vapor spectral bands (1.13 and 1.37  $\mu$ m) vary with COT (figure 3a) and  $IWV_{AC}$  (figures 3b and 3c). Simulations are performed with atmospheric profiles derived from the Air Force Geophysics Laboratory (AFGL) database (Anderson et al., 1986).

To perform this sensitivity study we introduced a set of one hundred different clouds by combining ten different cloud top heights  $(Z_t)$  ranging from 1 to 10 km in order to have a large variability of  $IWV_{AC}$  (from  $10^{-2}$  to  $25 \text{ kg.m}^{-2}$ ) and ten different cloud optical thicknesses (COT) ranging from 10 to 200.

Figure 3a shows the relationship between the radiance of the non-absorbing band according to the COT for two atmospheric profiles under the 1D cloud homogeneous assumption. The relation is monotonically increasing and non-depending on the

Figure 3. Simulated radiances of the  $C^3$  IEL 1.04  $\mu m$  band as a function of the COT (a), of the 1.13  $\mu m$  band (b) and 1.37  $\mu m$  band (c) as a function of  $IWV_{AC}$  for two atmospheric profiles from the AFGL database (Mid-Latitude Summer and Tropical profiles), for several cloud top heights ( $Z_t = 1, 2, 3, 4, 5, 6, 7, 8, 9,$  and 10 km), and various COT ranging from 10 to 200. In the figure (c), the rectangle represents a zoom of the figure for  $IWV_{AC}$  values between 0 and 5 kg·m<sup>-2</sup>. The Solar Zenith Angle (SZA) is set to 30° and the satellite observation angle (View Zenith Angle - VZA) is 0° (nadir).

water vapor absorption as excepted from the transmission curve shown in Figure 1. This band is thus necessary and useful to obtain information on the COT.

Figure 3b shows the results for these various cloud cases and exhibits, logically, a decrease in the simulated radiances as the  $IWV_{AC}$  increases with a decrease of the cloud top. Indeed, the greater the apparent radiation path through the atmosphere is,

the more water vapor the radiation interacts with, leading to a decrease in radiance. Radiance sensitivity is particularly high at lower water vapor contents but decreases as the water vapor content increases.

Figure 3c shows that radiance in the 1.37 μm spectral band becomes negligible when water vapor content exceeds 5 kg.m<sup>-2</sup>. It does not reach zero in the 1.13 μm band for values until 25 kg.m<sup>-2</sup>. The 1.37 μm band is thus mainly useful in retrieving  $IWV_{AC}$  for low water content above cloud either in the presence of high-level clouds or in a dry atmosphere. It is also worth noting the non-overlapping color curves, where each color corresponds to a specific COT value. The differences between the curves indicate a sensitivity of the two absorbing spectral bands to the 1D equivalent COT. This value is obtained using the information contained in the non-absorbing band centered at 1.04 μm. The hypothesis of a 1D homogeneous cloud assumption with a uniform extinction vertical profile is used and leads to errors in the  $IWC_{AC}$  retrieval as it will be discussed in section 5.2.4.

# 5 $IWV_{AC}$ retrieval





First, the retrieval algorithm presented in this paper is tested under idealized cloudy sky profiles using clear sky profiles from the AFGL database (Anderson et al., 1986), and then under realistic cloudy sky profiles, using atmospheric profiles provided by ECMWF-IFS database (https://www.nwpsaf.eu/site/software/atmospheric-profile-data/) to better evaluate its performance.

# 5.1 Using idealized cloudy profiles

This test set was built using the same approach as for the sensitivity test above, we artificially introduced a set of one hundred clouds in the AFGL database profiles, with a fixed cloud base height  $(Z_b)$  of 0.5 km and ten different cloud top heights ranging from 1 to 10 km. These artificial clouds are assumed to be composed entirely of water droplets with an effective radius of  $10 \, \mu m$  and with a COT varying from 10 to 200. The solar incidence angle is set to  $30^{\circ}$  and the satellite observation angle is  $0^{\circ}$  (nadir). The target  $IWV_{AC}$  values are derived from the AFGL tropical profile, while the first guess profile used to start the retrieval process is the SAS (Sub-Arctic Summer) profile, due to its smooth gradient from the top to the bottom of the atmosphere. Tests carried out with different first guess profiles, both in idealized and realistic scenarios, show that initial profile has an impact on in-cloud water vapor since we assume that RH = 100% between cloud base and cloud top altitudes. So, starting iterations with the smooth SAS profile leads to an underestimation of the in-cloud water vapor absorption as the SAS profile is drier than the tropical profile used to generate the test data.

As the retrieval of COT from non absorbing wavelength is well-known, figures related to the COT retrieval are not displayed here. Nevertheless, the developed retrieval algorithm allows to retrieve 1D equivalent COT with an average error less than 12% for COT values below 100 (less than 8% for values below 50). However, as it is well-known (e.g., Nakajima and King, 1990; Nakajima et al., 1991), if the 1D equivalent COT exceed 100, an accurate retrieval becomes difficult because of the asymptotic behavior of the radiances for large COT values (an average error of around 43% in this case). When COT is correctly retrieved, the in-cloud extinction profiles are identical in the simulated observations and in the model used for the retrieval. The differences in  $IWV_{AC}$  are thus explained mainly by the differences of in-cloud water vapor profiles. For optically thick

clouds COT is systematically underestimated leading to an underestimation of the in-cloud extinction coefficient. Although radiation penetrates less in optically thick clouds, the underestimation of the COT leads to more radiation interaction with in-cloud water vapor at the top of the cloud in the retrieval algorithm than in reality.




The  $IWV_{AC}$  retrieval is tested for two measurement vectors: the first one with only the 1.04 and the 1.13  $\mu$ m radiances and the other one with the three spectral bands centered at 1.04, 1.13 and 1.37  $\mu$ m. Absolute errors (retrieved minus target values) are displayed in figure 4.

Figure 4. Absolute errors obtained for the  $IWV_{AC}$  retrieval ( $IWV_{AC}$  retrieved minus  $IWV_{AC}$  target) as a function of the  $IWV_{AC}$  target values. Dotted lines represent results obtained using only the 1.04 and 1.13  $\mu$ m spectral bands, while full lines illustrate results using the three spectral bands (1.04, 1.13 and 1.37  $\mu$ m) in the measurement vectors. The colorbar represents the different COT values.

While both the 2-channel and 3-channel retrieval approaches show an increase in absolute error ( $IWV_{AC}$  retrieved minus  $IWV_{AC}$  target) with increasing water vapor content and decreasing COT, the 3-channel method exhibits a more consistent, nearly monotonic error growth with water vapor. In contrast, the 2-channel method performs reasonably well up to  $5 \text{ kg.m}^{-2}$ , but the error tends to flatten between 5– $10 \text{ kg.m}^{-2}$  and beyond. The absolute error is positive, indicating that the retrieved  $IWV_{AC}$  is overestimated due to less in-cloud water vapor absorption. This occurs because the water vapor profile within the cloud is not adjusted during the retrieval process, and the first-guess profile (AFGL SAS) is drier than the target AFGL tropical profile. Consequently, less radiation is absorbed within the cloud with the AFGL SAS model used for retrieval than with the AFGL tropical model used to simulate the test measurements. To compensate for this lower absorption and minimize the difference between the forward model simulations F(x) and the measurements y, the retrieved integrated water vapor above

the cloud is overestimated. For optically thick clouds, the same behavior appears to a lesser extent, since less radiation interacts with the in-cloud water vapor.

Tests made using the tropical profile as the first guess in order to have the same in-cloud water vapor absorption between the simulated measurements and the retrievals leads to smaller errors for thin clouds since the COT is well retrieved. Conversely, larger negative errors for low-level and optically thick clouds occur, as the algorithm underestimates the in-cloud extinction profile. Consequently, there is more in-cloud absorption and the algorithm underestimates water vapor above clouds. Based on these results, the use of the SAS profile as a first guess appears to introduce a compensatory effect that partially mitigates the systematic underestimation of large COT.

When considering only the simulated radiances at 1.04 and 1.13 µm (dotted lines in figure 4), absolute errors increases rapidly for low  $IWV_{AC}$  values (below  $5 \text{ kg.m}^{-2}$ ), to reach a peak above  $6 \text{ kg.m}^{-2}$  in the presence of optically thin clouds contrary to the retrieval with the three bands for which the absolute error remains less than  $0.5 \text{ kg.m}^{-2}$ . These results clearly shows that the highly absorbing band at 1.37 µm is essential to retrieve low water vapor content above clouds. Above  $5 \text{ kg.m}^{-2}$  the absolute errors values with two spectral bands remain roughly constant and get closer to the errors obtained using the three spectral bands as the 1.37 µm tends to be totally absorbed (radiances are close to 0).

When examining the combined information from the three spectral bands (solid lines), the absolute errors are significantly lower than  $1~\rm kg.m^{-2}$  for water vapor contents below  $10~\rm kg.m^{-2}$ , regardless of the optical thickness. As mentioned previously, as  $IWV_{AC}$  increases, the absolute errors also increase, particularly for thin clouds. In this case, we observe maximum absolute errors of about 3 to  $5~\rm kg.m^{-2}$  for cloud optical thickness (COT) values below 30. For optically thick clouds (COT > 80), the errors remain below  $2~\rm kg.m^{-2}$ , even for higher  $IWV_{AC}$  values. Overall, this figure highlights the necessity of using the three  $C^3IEL$  spectral bands to retrieve accurately the  $IWV_{AC}$ , and demonstrates good performance, especially for moderate to high optical thicknesses.

Note that C<sup>3</sup>IEL will not give information about the cloud base altitude, for the idealized case we use 0.5 km and a standard deviation obtained from the ECMWF-IFS database. In order to test the sensitivity of the proposed algorithm to this fixed parameter, we performed the retrievals by adding a large error on this parameter (2000 m).

Figure 5 shows these retrievals. Using the 3-channels approach, for cloud top altitude of 1 km, the algorithm do not converge. For low-level clouds with cloud top altitudes of 2 and 3 km, large absolute errors are observed (around  $-5 \text{ to } -2 \text{ kg.m}^{-2}$ , respectively for moderate COTs, purple to blue lines). Above these altitudes, for cloud tops from 4 to 10 km, this parameter does not significantly modify the results, and absolute errors are in the same range as previously.

### 5.2 Using realistic cloudy profiles







The ECMWF-IFS database provides realistic atmospheric cloud profiles at 137 pressure levels, including profiles of temperature, specific humidity, cloud liquid and ice water, along with fractional cloud cover. It also provides surface data such as pressure, temperature and albedo.

Figure 5. Same as figure 4 for the three spectral bands  $(1.04, 1.13 \text{ and } 1.37 \, \mu \text{m})$  configuration but with a large uncertainty of 2000 m for the cloud base altitude identified as a non-retrieved parameter.

# 5.2.1 Description of the realistic water vapor profiles




We restrict our analysis to profiles with clouds over the ocean to minimize surface effects (e.g., land, ice, snow), since the developed algorithm does not currently account for the underlying surface. Profiles at latitudes higher than 60° N/S are also excluded, since the  $C^3$ IEL mission will not observe at these latitudes because the stereo-restitution algorithm for the CLOUD imager (e.g., Dandini et al., 2022) is expected not to work well with low solar incidence, and also because convective clouds in development stage, not numerous at high latitude, are the main targets of the  $C^3$ IEL mission. Additionally, as the  $C^3$ IEL mission focuses on studying *cumulus congestus* and *cumulonimbus* clouds, clear sky profiles and those containing only *cirrus* clouds have been discarded from our study, as well as we choose to study single-layer clouds, excluding cases where a clear sky layer exists between two cloud layers. The same approach was applied for selecting high-level clouds, considering only continuous clouds with the condition that the cloud top height of the liquid phase ( $Z_{t,liq}$ ) must be greater or equal to the cloud base height of the ice phase ( $Z_{b,ice}$ ). Figure 6 shows the geographical distribution of the 232 selected profiles of low/mid level clouds and the 30 profiles corresponding to high-level clouds. For each profile, the  $IWV_{AC}$  is computed and is used as the target value, considered as the "truth". The distribution of these values are presented in figure 7.

For the ECMWF-IFS profiles, the cloud base and cloud top height are obtained from the Liquid/Ice Water Content (respectively LWC and IWC) profiles. We look for the first and last values of the LWC and IWC profiles above a threshold of  $10^{-3}$ 

**Figure 6.** Geographical distribution of the selected profiles from the ECMWF-IFS database used to test the developed retrieval algorithm. A total of 262 profiles, including 232 profiles with low/mid-level clouds liquid-only clouds (red points), and 30 profiles with high-level mixed-phase clouds (blue points) are selected.

 ${\rm g.m^{-3}}$ . The cloud phase is determined based on the LWC and IWC values, and the layers may be classified as liquid, ice or mixed phase. Then, in case of low/mid-level clouds,  $IWV_{AC}$  values range from 0.5 to  $21.3~{\rm kg.m^{-2}}$  but are predominantly below  $15~{\rm kg.m^{-2}}$ . Regarding high-level clouds, the values range from approximately  $0.06~{\rm kg.m^{-2}}$  to  $0.9~{\rm kg.m^{-2}}$ . Initially, the database did not include enough clouds at mid-high altitudes (6 - 10 km). To compensate for this deficiency, the cloud top height is artificially decreased by 4 km. As a result, the  $IWV_{AC}$  calculations for high-level clouds are derived from these adjusted profiles.

# 5.2.2 Results of the $IWV_{AC}$ retrieval: Low-/Mid-level clouds



Simulations are performed using three different Solar Zenith Angles ( $SZA=0^{\circ}$ ,  $30^{\circ}$ , and  $60^{\circ}$ ) and 11 Viewing Zenith Angles (VZA), based on the  $C^3$ IEL planned acquisition sequence, resulting in a total of 7,656 retrievals. As for retrievals under idealized conditions, the first guess profile is the SAS profile and the figures and analyses focus only on the  $IWV_{AC}$  retrieval. However, the algorithm developed also allows for the retrieval of the 1D equivalent cloud optical thickness (COT) and shows an average absolute error of -2 for COT < 100. In contrast, for COT > 100, the COT retrieval exhibits larger errors due to the asymptotic behavior of radiance as a function of COT, as for the idealized cases. Figure 8 illustrates the relationship between the retrieved and target values of the  $IWV_{AC}$  along the radiation path in the atmosphere (noted  $mIWV_{AC}$ ), where  $mIWV_{AC}$  is calculated as the  $IWV_{AC}$  multiplied by the air mass factor m, defined as:

Figure 7. Distribution of  $IWV_{AC}$  values calculated from the selected profiles. (a) target  $IWV_{AC}$  values for the 232 profiles containing low/mid-level clouds, (b) target values for the 30 profiles containing high-level clouds.

$$m = \frac{1}{\cos(SZA)} + \frac{1}{\cos(VZA)} \tag{7}$$

A good correlation is observed between the retrieved and target values, as indicated by a score  $(R^2)$  of 0.95 and the linear regression equation as a slope of 1.02 with a small systematic under-estimation of -0.73 kg.m<sup>-2</sup>. A convergence rate of 95.8% was achieved, indicating that 95.8% of the profiles and geometries successfully converge to a value above the threshold limit of 0.01 kg.m<sup>-2</sup>. The 4.2% of profiles that do not converge correspond to low and optically thin clouds, primarily clouds below 2 km and with a COT below 20, which is consistent with results obtained for idealized cases. The non convergence cases and the largest errors comes from the cloud base height  $(Z_b)$  errors and the way of computing to cloud extinction profile  $(C_{ext})$ , assumed to be vertically homogeneous and defined as the ratio of cloud optical thickness (COT) to its geometric thickness  $(Z_t-Z_b)$ . In this study, the cloud base height is defined as the average value of the ECMWF-IFS  $Z_b$  distribution, associated with an uncertainty of 45%, which represents, for low-level clouds, a large geometrical thickness uncertainty, impacting the cloud extinction coefficient value and the cloud vertical penetration. These results corroborate the conclusions of Albert et al. (2001) and the results obtained in section 5.1.




The error bars in Figure 8 represent the *a posteriori* uncertainty estimated by the retrieval algorithm. These errors comes from the errors measurement and the fixed (or non-retrieved) parameters. Although the results from the algorithm are satisfactory, only 84% of the retrievals and their associated errors capture the target value within the 3-sigma confidence interval. It highlights a limitation of the current developed algorithm to perfectly take into account the errors, *e.g.* coming from the forward model F(x). Since no information is available regarding the "true" extinction profiles in the measurements, the typical assumption of vertical uniformity is employed, and errors related to this assumption are not included in the error variance-

Figure 8. Relationship between the retrieved and target  $mIWV_{AC}$  values for profiles containing low-/mid-level clouds. The red line represents the linear regression and the black dashed line indicates the y = x line. Error bars represent the uncertainty estimated by the retrieval algorithm.

covariance matrix. Furthermore, after additional analysis, it was found that the unsatisfactory uncertainties correspond again to low-level clouds ( $Z_t < 2.5 \text{ km}$ ) and/or thin clouds (COT < 30). Regarding the statistics for the entire dataset, the absolute error distribution is shown in Figure 9a. Results show an average value of  $-0.18 \text{ kg.m}^{-2}$  and a standard deviation of  $0.92 \text{ kg.m}^{-2}$ , indicating a small underestimation of the retrieved  $IWV_{AC}$  in average. To further refine and better characterize the retrieval algorithm's behavior, RMSE values have been calculated according to cloud optical thickness (COT) and cloud top height ( $Z_t$ ) ranges as shown in Figure 9b.




As already seen for idealized cases, the RMSE values for  $IWV_{AC}$  are clearly higher for low-level and thin clouds, with values generally exceeding 1 kg.m<sup>-2</sup>. The RMSE decreases as  $Z_t$  and COT increase. RMSE values are below 0.8 kg.m<sup>-2</sup> for  $Z_t > 2,000$  m and COT > 50 and reach around 0.35 kg.m<sup>-2</sup> for  $Z_t > 3,000$  m and COT > 120. In optically thicker clouds, the penetration of radiation into the cloud-top layers is reduced, minimizing the impact of the extinction and water vapor profile assumptions used in the forward model. Additionally, as the cloud altitude increases from low to mid-level, the  $IWV_{AC}$  decreases rapidly, enhancing the sensitivity of the 1.37 µm band, as demonstrated in the idealized cases (Figure 4).

Figure 9. a) Distribution of the absolute error (retrieved minus target) for the whole dataset, and b) RMSEs calculated according to various  $Z_t$ , and COT ranges.

# 5.2.3 Results of the $IWV_{AC}$ retrieval: High-level clouds



As for low-/mid-level clouds, retrievals have been calculated for the same 33 geometries, which lead to a total of 990 retrievals. Results for  $mIWV_{AC}$  are presented in Figure 10. In comparison with Figure 8, the convergence rate for high-level clouds is now 100%. This is due to the absence of low-level or optically thin clouds in the selected profiles. For this type of clouds, COT values exceed 100 allowing the retrieval algorithm to converge for every profile. In terms of COT retrieval, the errors become considerable as beyond an optical thickness of 100, the spectral band sensitivity at 1.04  $\mu$ m is not sufficient to have a reliable retrieval under the assumption of an infinite cloud. The extinction profile in the cloud model used for the retrieval is thus impacted accordingly. Regarding the  $mIWV_{AC}$  retrieval, the algorithm tends to overestimate the  $mIWV_{AC}$  retrieved value for most of the cases with a linear regression slope of 1.49 and a relatively important dispersion as the  $R^2$  score is 0.54. These less favorable results, compared to the low- and mid-level cloud cases, are primarily due to six profiles (198 retrievals) where large discrepancies exist between the observed extinction profiles and those assumed in the retrieval algorithm, as discussed in the following section. Overall 76% of the retrieved  $mIWV_{AC}$  values and their uncertainties capture the target values for high-levels clouds (within the 3-sigma confidence interval), with a global  $IWV_{AC}$  RMSE of 0.33 kg.m<sup>-2</sup>.

Figure 10. Same as Figure 8 but for profiles containing high-level clouds.

# 5.2.4 Discussion concerning the underestimation or overestimation of the retrieved $IWV_{AC}$

As stated in the Figure 10 comments, there is a quasi systematic overestimation of the retrieved  $IWV_{AC}$  value for the mixed-phase clouds (*i.e.*, high-level clouds) while there is a small underestimation in the case of liquid clouds (*i.e.*, low-/mid-level clouds, Figure 9a).

Figure 11. Absolute humidity and extinction profiles derived from the ECMWF-IFS database and retrieved from the developed retrieval algorithm. Three cases are showed to illustrate the underestimation of  $IWV_{AC}$  for liquid cloud cases in (a), the overestimation of  $IWV_{AC}$  for the mixed-phase clouds in (b) and (c). The dashed black line indicates the "true" water vapor profile and the solid red line, the water vapor profile obtained at the last iteration of the OEM. The dashed green curve represents the "true" cloud extinction profile while the solid green line corresponds to the cloud model extinction profile.  $Z_b$  and  $Z_t$  are represented by horizontal yellow lines.

We identified three cases that explain the underestimation, Figure 11a, or overestimation, Figure 11b and Figure 11c, of the water vapor content above the clouds. In this figure, realistic absolute humidity (black dashed lines) and extinction profiles (green dashed line) within the cloud (obtained from the ECMWF-IFS database) are plotted alongside profiles retrieved with the forward model assumptions. The dark green lines represent vertically uniform cloud extinction profiles, while the solid red lines depict water vapor profiles, assuming 100% relative humidity within the cloud for the SAS profile. The cloud top height ( $Z_t$ ) and cloud base height ( $Z_b$ ) are delimited by yellow horizontal lines.

In Figure 11a, the extinction coefficient in the upper cloud layers is lower in the model compared to the "true" extinction profile. This leads to deeper vertical penetration of radiation into the cloud compared to the realistic profile, resulting to increase the absorption by water vapor within the cloud. To compensate for this effect, the retrieval algorithm reduces absorption above the cloud resulting in lower  $IWV_{AC}$  (8.50 vs 9.71 kg.m<sup>-2</sup>) in order to minimize the difference between the measured (y in Eq. (1)) and simulated radiances F(x,b).

In Figure 11b, although the algorithm tends to underestimate the total COT, it overestimates the extinction profile in the upper layers (above 7 km). This overestimation reduces the cloud penetration and, consequently, the in-cloud absorption. To compensate this effect the algorithm overestimates the  $IWV_{AC}$  (0.20 vs 0.06 kg.m<sup>-2</sup>). Additionally, the in-cloud water vapor profile (dashed black line) is higher in the realistic profile. When combined with the greater radiation penetration, this amplifies the absorption difference between the target and the retrieved value, further contributing to the overestimation.

Figure 11c shows a second example that leads to an overestimation of the retrieved  $IWV_{AC}$ . In this case, the algorithm underestimates the retrieved COT but the extinction profile within the upper part of the cloud is very similar to the observed one, which means that cloud penetration between the observation and the retrieval is consistent. The main difference lies in the in-cloud humidity profile which is largely underestimated in the model, leading to less in-cloud absorption and thus an overestimation of the  $IWV_{AC}$ .

The combined effects of under/overestimation of the top layers extinction coefficient and the under/overestimation of the in-cloud humidity are certainly the main source of errors in the current algorithm. Note that these effects can sometimes counterbalance each other, for example in case of less vertical penetration into the cloud associated to higher absolute humidity values.

### 6 Conclusions






The objective of this study is to show the feasibility of estimating the integrated water vapor content above cloud ( $IWV_{AC}$ ) and the cloud optical thickness (COT), in the context of a future space mission project named  $C^3IEL$  (Cluster for Cloud evolution, ClImatE, and Lightning) in order to develop a retrieval algorithm, tested above ocean surfaces to minimize the effect of the surface, and it excluded high latitudes, above  $60^{\circ}$  N/S as the  $C^3IEL$  instruments will not acquire measurements at these latitudes. This mission aims at investigating the development of convective clouds with a high spatial and temporal resolution, the electrical activity associated with these clouds, and the water vapor content above and around cloud. The developed algorithm is based on a Bayesian probabilistic approach, the Optimal Estimation Method (OEM), which is an

iterative fitting method. It is similar to the least square method but it offers additional possibilities as it permits to consider not only the measurement itself but also supplementary information (fixed parameters, *a priori* knowledge, *etc.*) as well as associated uncertainties for the estimation of several variables.

A study has been conducted to evaluate if the three SWIR spectral bands of the  $C^3$ IEL mission can be used to retrieve the integrated water vapor content above cloud ( $IWV_{AC}$ ). First, using simulations in the two water vapor spectral bands (1.13 and 1.37  $\mu$ m), we show that radiance values decreases as  $IWV_{AC}$  increases because of the water vapor absorption. We note that the simulated radiances in the highly absorbing band (1.37  $\mu$ m) tends toward 0 for  $IWV_{AC}$  values greater than 5 kg.m<sup>-2</sup>, *i.e.*, emphasizing the fact that this band is mainly useful for low  $IWV_{AC}$ , in dry atmosphere or above high clouds.







The developed algorithm is then first tested under idealized cloudy sky conditions with homogeneous cloud extinction profiles as assumed in the inversion model. The absolute errors of  $IWV_{AC}$  tends to increase as  $IWV_{AC}$  increases and COT decreases, thus the vertical penetration in the cloud. We also demonstrate the advantages of combining the two water vapor spectral bands, specifically, for the low water vapor content values where the absolute errors when combining the two spectral bands remains below  $0.5 \text{ kg.m}^{-2}$ , regardless of the cloud optical thickness.

Then, the algorithm was tested on realistic profiles obtained from the ECMWF-IFS database. We focused on profiles featuring exclusively cloudy sky conditions over the ocean and within a latitude range of  $60^{\circ}$  N/S. We identified 232 cloudy profiles containing low-/mid-level clouds and 30 profiles containing high-level clouds (excluding cirrus). The algorithm exhibited a 95.8% convergence rate for realistic profiles with low-/mid-level clouds. The  $IWV_{AC}$  results show a good correlation between the retrieved values and the target values ( $R^2$  of 0.95 and a slope of 1.02). The RMSE is below 1 kg.m<sup>-2</sup> for COT values above 50 and  $Z_t > 2,000$  m. The RMSE tends to increase for low clouds and low cloud optical thickness. In case of high clouds, the  $IWV_{AC}$  values are low so more difficult to retrieve. The RMSE is however still less than 1 kg.m<sup>-2</sup>. The linear regression slope is 1.49 and the score is 0.54. The analysis shows that the errors comes mainly from the extinction profiles and water vapor profiles assumed in the inversion model.

The algorithm presented in this paper is a preliminary one and could be improved. This first version of the algorithm is currently over ocean with a constant surface albedo. In future versions, more realistic surfaces could be implemented using wind speed and the Cox and Munk (1954a, b) model above ocean, and surface albedo above continents coming from ancillary data such as ECMWF-IFS data for wind speed and MODIS or Sentinel-2 products for surface albedo values. Another way can be to use a more typical and realistic cloud extinction profiles in both liquid and mixed-phase clouds. This non-uniform profile can use a simple parametrization (*e.g.*, Matar et al., 2023) or be based on radar measurements (*e.g.*, Carbajal Henken et al., 2014). This improvement could possibly reduce the bias. Once the definition of the vertical profile in the algorithm has been improved, one way to test it with realistic convective cloud profiles could be to use LES simulations or the EarthCARE reconstruction (Barker et al., 2011). The cloud model phase used during the retrieval process can also be improved, particularly the assumption of 4 km made for the top of the liquid phase, for tests on mixed-phase clouds, given that supercooled liquid clouds can be present well above 4 km altitude. A better definition of the different cloud phases would allow for a better representation of the cloud's optical properties. Another way relies on the fixed parameters accuracy of cloud base height and effective radius, which both significantly affect cloud extinction values and difference in cloud vertical penetration. To improve the determination of

these values, climatological data of  $Z_b$  or  $R_e$ , specific to the geographical location of the measurements could be used. In the same way, temperature profile climatology could be used to better constrain the RH=100% assumption in the cloud model. Furthermore, research works have to be done to exploit the multi-views of the  $C^3$ IEL mission. The use of two simultaneous view angles should help reduce the uncertainties by better constraining the retrieval. However, using acquisitions spaced 20s apart requires the development of a more complex algorithm, as the retrieved cloud top height is expected to vary sufficiently between two successive acquisitions (e.g., Dandini et al., 2022). Moreover, at the observation scale of approximately 125 m, 3D radiative transfer can introduce effects such as radiative smoothing (e.g., Marshak, 1995; Davis et al., 1997), as well as illumination and shadowing Várnai (2000) that roughen the radiative fields. These factors influence shortwave radiances and may impact the retrieval of water vapor content above clouds even if 3D radiative effects are anticipated to impact both absorbing and non-absorbing bands similarly due to their close spectral proximity. However, in addition for tilted view geometry of a finite cloud, the cloud base may appears significantly higher in the atmosphere than assumed in the cloud model, which could impact retrieval accuracy. To assess the assumption of a flat, homogeneous, and infinite cloud, more realistic cloud representations using 3D radiative simulations would be required to compute the measurements. This is a significant task beyond the scope of this paper, which aims to explore, as a first step, the possibility of retrieving integrated water vapor above clouds.

## 450 Appendix A: Variables from the *a posteriori* variance-covariance matrix

In our study, the *a priori* variance-covariance matrix (see Eq. (4)) is a diagonal matrix where the diagonal terms are the squared standard deviations for each parameter contained in the *a priori* state vector:

$$S_a = \begin{bmatrix} \sigma_{COT}^2 & 0\\ 0 & \sigma_{IWV_{AC}}^2 \end{bmatrix} \tag{A1}$$

with  $\sigma_{COT}$  and  $\sigma_{IWV_{AC}}$  obtained by multiplying the *a priori* state vector  $x_a$  by the arbitrarily chosen *a priori* error.

$S_{\epsilon}$ , the error variance-covariance matrix is expressed, in our study, as follows:

$$S_{\epsilon} = S_y + S_{fp} \tag{A2}$$

 $S_y$  is the measurement variance-covariance matrix, and  $S_{fp}$  describes the variance-covariance matrix linked to the fixed parameters:

$$S_y = \begin{bmatrix} \sigma_{y1}^2 & 0 & 0\\ 0 & \sigma_{y2}^2 & 0\\ 0 & 0 & \sigma_{y3}^2 \end{bmatrix}$$
(A3)

and,

$$S_{fp} = K_b S_b K_b^T \tag{A4}$$

with,

$$K_b = \frac{dF(b)}{db} \bigg|_x = \frac{F(b+db) - F(b)}{db} \bigg|_x \tag{A5}$$

where b represent the fixed parameter (see table 1) and db is equal to 1% of the fixed parameter value.  $S_b$  is a diagonal matrix whose diagonal elements represent the squared standard deviation associated with the different fixed parameters of the forward model:

$$S_b = \begin{bmatrix} \sigma_{Albedo}^2 & 0 & 0 & 0 & 0 \\ 0 & \sigma_{Z_b}^2 & 0 & 0 & 0 \\ 0 & 0 & \sigma_{Z_t}^2 & 0 & 0 \\ 0 & 0 & 0 & \sigma_{R_{e,droplet}}^2 & 0 \\ 0 & 0 & 0 & 0 & \sigma_{R_{e,droplet}}^2 \end{bmatrix}$$

$$(A6)$$

The different elements of this matrix are obtained by calculating the product of the relevant fixed parameter with the uncertainty attributed to it (see table 1).

- Author contributions. Each author has played a significant role in the development and scientific exploration presented in this study. RP and GP performed the study, developed the algorithm and write the original draft of the article. CC provided support for radiative transfer simulations and the analysis of corresponding data and retrieval results. OP contributed to the analysis of the atmospheric database (ECMWF-IFS) and the interpretation of retrieval results as well. CP offered expertise in the optimal estimation method and contributed to the analysis of radiative transfer simulations and retrieval outcomes.
- 475 Competing interests. There is no competing interests.

Acknowledgements. We are grateful to the CNES/TOSCA program which has funded this work as part of the preparation of the Cluster for Cloud evolution, Climate and Lightning French-Israeli space mission project. We acknowledge the AERIS/ICARE data center for providing the ARTDECO radiative transfer code free of charge. We would also like to thank L-C. Labonnote for his precious help and expertise on the ARTDECO tool and F. Thieuleux for his precious help during the development of the retrieval algorithm.

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
