# Peer review of "Water Vapor Content Retrieval Under Cloudy Sky Conditions from SWIR Satellite Measurements in the Context of C3IEL Space Mission Project"

_EGUsphere, 2025_

## Author Comment (AC1)

**General comment from the reviewer**

"This paper addresses the long standing (and still of paramount importance) topic of convective cloud formation/development and the surrounding water vapour (WV) conditions that lead to it.

The problem of WV retrieval above clouds, although attempted in the literature (as mentioned by the authors), remains very appealing and very topical nowadays, because no systematic/operational retrieval exists at the moment in Meteorological agencies. The dedicated WV imager on the C3IL mission will cover this gap, allowing a 3-year long coverage.

The problem of WV retrieval above clouds is successfully addressed by proposing an optimal estimation approach, using SWIR imaging measurements in 3 relevant channels (covering the spectral range from  $\sim$ 1 to 1.3 $\mu$ m) in and off the water vapour absorption band and taking into account the relevant factors affecting (together with the water vapour profile) the observed radiances in these channels, i.e., the surface properties such as albedo and the cloud optical thickness (COT) and height).

The retrieval approach is motivated based on previous works conducted with POLDER and MERIS data and demonstrated using appropriate simulated data, and test retrieval using both idealized and realistic atmospheric profiles. The results are convincing (i.e., the proposed algorithm is clearly sensitive to the water vapour amount above clouds, and its quantification is reliable within well motivated uncertainty (i.e. impact of realistic cloud profiles, of high COT, of low WV content, etc.).

Although demonstrated on simulated data only, the method opens interesting perspectives for the 3-year limited C3IL mission, and maybe for possible longer future satellite missions equipped with the same channels as the C3IL/WV imager.

All this considered, I have no doubt in recommending the publication of this paper after minor revision. In the attached PDF, I added all my comments to the text. Most of them are typos corrections or request for clarifications (e.g., clarify in abstract/conclusion that the proposed algorithm is demonstrated only over ocean and excludes latitudes higher the ±60 deg). However, I also added suggestions and comments that in my opinion may further improve the quality of this work (e.g., future work using real profiles and 3D cloud reconstruction from EarthCARE). The authors shall go through them, providing feedback where required."

**Dear Loredana Spezzi,**

Thanks for your comments and the suggestions provided. In the revised manuscript, all the typos and suggested reformulations are accounted for. We have listed your comments that need an answer below (in red), explain, when necessary, how we have addressed each point in the revised manuscript, and provided new sentences (in green).

**Comment on page 1 (old file):**

"Please clearly state here that the retrieval proposed in this paper is demonstrated only over ocean and excludes latitudes higher the ± 60 deg."

**New sentence (new file, page.1 lines.1-3):**

"A retrieval algorithm of integrated water vapor content above cloud, using shortwave infrared observations, is developed and evaluated through idealized and realistic atmospheric profiles, with its application currently limited to oceanic regions and latitudes within  $\pm$  60 degrees."

**Comment on page 2 (old file):**

**Do you mean "depending on the temperature"?**

Yes, we'd like to say that at a given temperature, saturation is reached. We modify this sentence. New sentence (new file, page.2 lines.39-40):

"Indeed, at a given temperature, saturation is reached when the water vapor partial pressure equals the saturation vapor pressure."

**Comment on page 4 (old file):**

"Please specify that the C3IEL mission is still under development, launch is expected in 2027 and the duration is at least 3 years according to WMO OSCAR, see <a href="https://space.oscar.wmo.int/satellites/view/c3iel">https://space.oscar.wmo.int/satellites/view/c3iel</a> b"

This is now clearly stated with an update of the expected launch date by adding this sentence (new file, page.4 lines.95-97):

"The mission is under development and is scheduled to be launched at the end of 2028, with a minimum expected duration of two years that could be extended to three years."

**Comment on page 5 (old file):**

"Please clarify here in the caption or in the text why these tophat rectangular filters are shown and used for the sensitivity study. Are the filters SRFs not yet defined?"

Indeed, at the time of this study and still now, the SRFs are not yet defined. Consequently, the choice was made to use rectangular spectral response function. We add this information in the caption (new file, page.5 Figure 2. caption):

"Note that rectangular spectral response functions are used, as the actual SRF are not known."

**Comment on page 7 (old file):**

"This is just a suggestion for improvement. Instead of assuming a constant surface albedo over ocean, it could be easily calculated case by case (i.e. for the location and profiles shown in Figure 5) using the ECMWF wind speed and the Cox and Muck 1954 approach."

Thank you for your good suggestion. We've added a paragraph in the conclusion and perspective section regarding the account for more realistic surface albedo in the next version of the algorithm. New paragraph (new file, page.21 lines.423-426):

"This first version of the algorithm is currently over ocean with a constant surface albedo. In future versions, more realistic surfaces could be implemented using wind speed and the Cox and Munk (1954a, b) model above ocean, and surface albedo above continents coming from ancillary data such as ECMWF-IFS data for wind speed and MODIS or Sentinel-2 products for surface albedo values."

**Comment on page 8 (old file):**

"I could not find in the text the assumed the first guess values for COT and IWV\_ac. Can you please specify them? Are they the same in all test retrievals?"

You are right, we missed to indicate these values here. We've added this sentence (new file, page.8 lines.188-190):

"For the COT, the first guess value is 10 and for the IWV\_AC, the first guess value is calculated by integrating the SAS (Sub-Arctic Summer) water vapor profile from the cloud top altitude up to 20 km."

**Comment on page 9 (old file):**

"Are the radiance measurements expected from the C2IL/WV image precise enough to appreciate this small differences? In the state vector "x" you have both COT and IWV\_AC, and you are assuming an uncertainty of 5% in the measurement vector Y (see Sect. 3.1). Is this enough to allow the retrieval to distinguish among the COT in curves? Please clarify both points in the text."

The C3IEL WV imagers should provide radiance data with an accuracy better than 5% (random error at 1 sigma). This is why we consider an uncertainty of 5% in the measurement vector Y. Figure 3 are sensitivity curves and corresponds to simulations without accounting for the measurement error. So, you are right, it is not obvious that the random error on the measurement, linked to instrument precision, allows to distinguish among the COT in curves. To consider the limited accuracy of the measurements, we realized the retrieval described in section 5, and which consider the measurement noise (through 5% uncertainty on measurement vector). Results regarding the ability to measure COT despite the noise measurement are given in lines 238 to 241 (new file, page.10) and show that 5% accuracy is enough. We made the following changes in the text (new file, page.6 lines.141-143):

"Measurement vector data are associated with an uncertainty of 5%. This value corresponds to the requirement on the accuracy of the radiance measured by the instrument (random error at 1 sigma), corresponding to random noise."

We also rephrased and completed this sentence (new file, page. 10 lines. 217-221):

"The differences between the curves indicate a sensitivity of the two absorbing spectral bands to the 1D equivalent COT. This value is obtained using the information contained in the non-absorbing band centered at 1.04 µm. The

hypothesis of a 1D homogeneous cloud assumption with a uniform extinction vertical profile is used and leads to errors in the IWC\_AC retrieval as it will be discussed in section 5.2.4."

We also completed the analysis of the results, section 5.1, by adding the following paragraph (new file, pages.10-11 Lines.241-246):

"When COT is correctly retrieved, the in-cloud extinction profiles are identical in the simulated observations and in the model used for the retrieval. The differences in IWV\_AC are thus explained mainly by the differences of in-cloud water vapor profiles. For optically thick clouds COT is systematically underestimated leading to an underestimation of the in-cloud extinction coefficient. Although radiation penetrates less in optically thick clouds, the underestimation of the COT leads to more radiation interaction with in-cloud water vapor at the top of the cloud in the retrieval algorithm than in reality."

**Comment on page 10 (old file):**

"Moreover, please note that your nonabsorbing channel centered at 1.04  $\mu m$  is not one the standard ones used to retrieve the COT (normally 0.6 or 0.8 microns are used), so you may consider adding a figure showing the relationship between the 1.04  $\mu m$  non-absorbing channel and COT."

Our first idea was to not put this plot in the paper as it is similar to the well-known one with visible channels. However, following your suggestion, we add it as a third subplot in Figure 3, modify the title of section 4, adjust the introductory paragraph of this section, and add a sentence to comment on the new subplot.

New section 4 title (new file, page.8 line.199):

"Sensitivity of the three spectral bands to COT and IWV\_AC"

Adjustment of the introductory paragraph (new file, page. 8 lines. 200-201):

"In this section, we examine how the radiances simulated in the non-absorbing band (1.04  $\mu$ m) and in the water vapor spectral bands (1.13 and 1.37  $\mu$ m) vary with COT (figure 3a) and IWV\_AC (figures 3b and 3c)."

New Figure 3 caption (new file, page.9 Figure 3. caption):

"Simulated radiances of the C3IEL 1.04  $\mu$ m band as a function of the COT (a), of the 1.13  $\mu$ m band (b) and 1.37  $\mu$ m band (c) as a function of IW VAC for two atmospheric

profiles from the AFGL database (Mid-Latitude Summer and Tropical profiles), for several cloud top heights (Zt = 1, 2, 3, 4, 5, 6, 7, 8, 9, and 10 km), and various COT ranging from 10 to 200."

**Added sentence (new file, pages.8-9 lines.206-209):**

"Figure 3a shows the relationship between the radiance of the non-absorbing band according to the COT for two atmospheric profiles under the 1D cloud homogeneous assumption. The relation is monotonically increasing and non-depending on the water vapor absorption as excepted from the transmission curve shown in Figure 1. This band is thus necessary and useful to obtain information on the COT."

**Subplot added to Figure 3 (new file, page. 9 new subplot a):**

"I would say that the 3 channels method is better behaved, because the error increases almost monotonically with the WV. When using the 2 channels approach, the error behavior is ok until 5 kg/m2 but the flattens between 5-10 kg/m2 and above."

**New sentence (new file, page.11 lines.250-253):**

"While both the 2-channel and 3-channel retrieval approaches show an increase in absolute error (IWV\_AC retrieved minus IWV\_AC target) with increasing water vapor content and decreasing COT, the 3-channel method exhibits a more consistent, nearly monotonic error growth with water vapor. In contrast, the 2-channel method

performs reasonably well up to 5 kg.m–2, but the error tends to flatten between 5–10 kg.m–2 and beyond."

**Comment on page 11 (old file):**

"It is clear to me why you concentrate over ocean given the complex behavior of the albedo over land and especially high contribution over snow/ice covered surfaces (which indeed you encounter especially at high latitudes). However, C3IL is on a polar orbit, so it observed high latitudes and polar areas. Thus, I think you should give in the text these explanations/motivations that lead you to exclude land and high latitudes."

Indeed, for simplicity, this first version of the algorithm was only developed for cloudy scenes over ocean but extension to scene over land is planned for the future. Concerning the observation at high latitudes, no sequence of acquisitions will be programmed first because the stereo-restitution algorithm for the CLOUD imager is excepted not to work well with low solar incidence and, also because convective clouds in development stage, not numerous at high latitude, are the main targets of the C3IEL mission. We have rephrased the first sentence of section 5.2.1 (new file, page.13 lines.292-296):

"We restrict our analysis to profiles with clouds over the ocean to minimize surface effects (e.g., land, ice, snow), since the developed algorithm does not currently account for the underlying surface. Profiles at latitudes higher than 60° N/S are also excluded, since the C³IEL mission will not observe at these latitudes because the stereo-restitution algorithm for the CLOUD imager (e.g., Dandini et al., 2022) is expected not to work well with low solar incidence, and also because convective clouds in development stage, not numerous at high latitude, are the main targets of the C3IEL mission."

Moreover, we added this information to the conclusion section as mentioned above (new file, page.21 lines.423-426). Then, we added complementary information in section 2 (new file, page.4 lines.99-100):

"Indeed, the satellites will rotate to track the same scene for 200 seconds and capture a sequence of 11 acquisitions of two simultaneous observations."

**And (new file, page.4 lines.102-106):**

"The viewing angles for each satellite will be approximately ±50°, -42°, -32°, -20°, and -7° on each side of the observed scene. Additionally, the first satellite will include a -55° angle, while the second satellite will include a +55° angle. Between each sequence of acquisitions, the satellites will have to return to their initial attitude, which implies a limited number of 4 sequences per orbit. The latitudes of these observation sequences will be chosen according to climatology of convective clouds."

**Comment on page 12 (old file):**

"What do you mean by "initially"? Did you then try with more mid-high profiles? I would say it is better to try again with realist profiles rather than artificially modified."

The ECMWF-IFS database doesn't have many profiles with cloud top at intermediate levels. For test purposes, we add artificially to the database these intermediate clouds by decreasing the cloud top value of high cloud by 4km. We agree with your comment on the use of more realistic profiles, which will be the subject of future developments and improvements as stated in the conclusion (new file, page.21 lines.429-431):

"Once the definition of the vertical profile in the algorithm has been improved, one way to test it with realistic convective cloud profiles could be to use LES simulations or the EarthCARE reconstruction (Barker et al., 2011)"

**Comment on page 17 (old file):**

"Same as in the abstract, please remind here to the reader that the algorithm is proven over ocean only and high latitudes are excluded."

We added the following sentence in the conclusion section (new file, page.20 lines.396-398):

"tested above ocean surfaces to minimize the effect of the surface, and it excluded high latitudes, above 60° N/S as the C3IEL instruments will not acquire measurements at these latitudes."

**Comment on page 18 (old file):**

I think it is worth to mention that the proposed algorithm can be possibly proven using realistic profiles from EarthCARE. Of course this would required extra simulation of C3IL/WV imaging measurements matching the date/time of EarthCARE observations. EarthCARE offers also a 3D cloud-reconstruction product that should be available by end 2025

(https://rmets.onlinelibrary.wiley.com/doi/full/10.1002/qj.824).

Thank you for your comment, indeed EarthCARE is a very good candidate to evaluate the algorithm and the upcoming improvements. We added this sentence in the perspective section (new file, page.21 lines.429-431):

"Once the definition of the vertical profile in the algorithm has been improved, one way to test it with realistic profiles of convective cloud can be to use LES simulations or the EarthCARE reconstruction (Barker et al. 2011).

"Will the cloud height vary so much in 20 sec? Maybe in certain condition of high wind speed and vorticity, but not in all conditions."

C3IEL is dedicated to convective clouds during their development stage (before the anvil formation) with one of the main objectives being to retrieve the cloud vertical development we expect (and hope) that the cloud top altitude will vary sufficiently between two acquisitions. LES simulations used to develop the CLOUD algorithm (Dandini et al. 2022) show cloud top altitude increase of 3-5 m/s which leads to 60-100 meters in 20s and 600 to 1000 meters in 20os. We have reformulated it in the perspective section and added the reference below (new file, page.22 lines.439-441):

"However, using acquisitions spaced 20s apart requires the development of a more complex algorithm, as the retrieved cloud top height is expected to vary sufficiently between two successive acquisitions (e.g., Dandini et al., 2022)."

**New references (new file, page 24):**

Barker HW, Jerg MP, Wehr T, Kato S, Donovan DP, Hogan RJ. 2011. A 3D cloud-construction algorithm for the EarthCARE satellite mission. Q. J. R. Meteorol. Soc. 137: 1042–1058. DOI:10.1002/qj.824

Cox, C. and Munk, W.: Statistics Of The Sea Surface Derived From Sun Glitter, Journal Of Marine Research, pp. 198–227, <a href="https://elischolar.495library.yale.edu/journal-of-marine-research/814">https://elischolar.495library.yale.edu/journal-of-marine-research/814</a>, 1954a.

Cox, C. and Munk, W.: Measurement of the Roughness of the Sea Surface from Photographs of the Sun's Glitter, Journal of the Optical Society of America, p. 838, <a href="https://doi.org/10.1364/JOSA.44.000838">https://doi.org/10.1364/JOSA.44.000838</a>, 1954b.

Dandini, P., Cornet, C., Binet, R., Fenouil, L., Holodovsky, V., Y. Schechner, Y., Ricard, D., and Rosenfeld, D.: 3D cloud envelope and cloud development velocity from simulated CLOUD (C3IEL) stereo images, Atmos. Meas. Tech., 15, 6221–6242, https://doi.org/10.5194/amt-15-6221-2022, 2022.

---

## Author Comment (AC2)

**General comment from the reviewer**

"This manuscript describes a retrieval algorithm of integrated water vapor content above cloud form shortwave infrared observations as will be measured with the C3IEL mission. The manuscript is of interest for AMT. However, I recommend some clarifications to be added to the manuscript, related to the questions below. Please address these issues in the revised manuscript."

https://doi.org/10.5194/egusphere-2025-787-RC2

**Dear reviewer.**

Thanks for your comments and your questions. In the revised manuscript all the typos and suggested reformulations are accounted for, and your questions have been addressed. We have listed your questions below (in red), then explained how we have addressed each point in the revised manuscript and provided new sentences (in green).

"The relative humidity is assumed to be 100% in cloud. I assume here the relative humidity is defined with respect to liquid water. This assumption is then true for water clouds, but not necessarily true for ice clouds. What is the impact of this assumption for ice clouds?"

The in-cloud relative humidity was set to 100% with respect to liquid water for all retrievals. This is a valid assumption for liquid-phase clouds, but it does lead to an overestimation of in-cloud humidity for ice clouds (saturation vapor pressure over ice is lower). However, we can expect this assumption to have a limited impact on the results. The cloud optical thickness constrains the depth to which radiation can penetrate. In optically thick clouds (which is the case for deep convective cloud), the radiation reaching the instrument comes primarily from above the cloud top and the uppermost layers of the cloud. So, the absorption occurring deeper within the cloud, and the influence of the in-cloud humidity profile should remain low. However, to make sure that this assumption is correct we have made the test, and the results are very similar. The histogram representing the difference (retrieved mIWV\_AC using RH=100% over ICE minus retrieved mIWV\_AC using RH=100% over Liquid) is presented below. As expected, the influence is low (mainly less than 0.2 kg/m2) and as there is less in-cloud water vapor using the calculation over ice, the algorithm compensates this by adding water vapor above the cloud (hence the quasi-systematic positive difference).

Though this test confirmed our hypothesis, we acknowledge that RH\_ice should be used in the retrieval algorithm for high level clouds, so we have modified the corresponding plots in the paper. We have added a sentence clarifying this point in the manuscript (new file, page.8 lines. 183-185):

"For low- and mid-level liquid clouds (Section 5.2.2), the assumption is applied with respect to liquid water. For high-level mixed-phase clouds (Section 5.2.3), relative humidity is defined as 100% with respect to liquid water below 4 km and with respect to ice above."

"For clouds with tops higher than 4 km, cloud tops are assumed to be ice. Many supercooled liquid clouds extend much higher than 4 km. What is the impact of a wrong cloud top phase assumption to the results?"

This is a valid concern, and we agree that supercooled liquid clouds can be present well above 4 km altitude, especially in convective systems. However, we have no information to know the exact vertical profile inside the cloud, so we have to make an assumption. The 4 km threshold used in our study is based on the average "0°C isotherm" level from the ECMWF-IFS profiles. An incorrect cloud phase assumption could affect the cloud optical properties, especially the single scattering albedo, asymmetry factor, and extinction coefficient, which could influence the estimation of cloud transmittance and the retrieved integrated water vapor content above cloud. In the future, we plan to improve the cloud model defined in our retrieval algorithm and use a more realistic and complex cloud vertical structure. We have added a sentence in the conclusion and perspective section (new file, page.21 lines. 431-434):

"The cloud model phase used during the retrieval process can also be improved, particularly the assumption of 4 km made for the top of the liquid phase, for tests on mixed-phase clouds, given that supercooled liquid clouds can be present well above 4 km altitude. A better definition of the different cloud phases would allow for a better representation of the cloud's optical properties."

"What is the sensitivity of the results to the assumed base height? The sensitivity is briefly discussed in section 5.2.2., but I think it should be systematically investigated related to the analysis shown in figures 3 and 4."

Thanks to this comment, we realized that there was a mistake in the value of the cloud base used in the idealized cases (section 5.1). We modified this value to its actual value (new file, page.10 line.228): "0.5 km".

Tests were carried out to see the effect of the cloud base altitude on the retrieval in the idealized cases where we increased the error on this fixed parameter from 320m (in the current version) up to 2000m. Results are presented in the figure below (using the three channels for the retrievals), added to the document (new file, page.13 Figure 5.).

Caption (new file, page.13 Figure 5.): Same as figure 4 for the three spectral bands (1.04, 1.13 and 1.37  $\mu$ m) configuration but with a large uncertainty of 2000 m for the cloud base altitude identified as a non-retrieved parameter.

As can be seen, the retrievals for low-level clouds with CTH below 4 km are largely underestimated, especially for optically thin clouds, while retrievals with CBH = 1 km are not possible (no convergence for all considered COTs), whereas for clouds with tops from 4 km, the impact is not significant. The following paragraph has been added to the manuscript (new file, page.12 lines.280-286):

"Note that C3IEL will not give information about the cloud base altitude, for the idealized case, we use 0.5 km and a standard deviation obtained from the ECMWF-IFS database. In order to test the sensitivity of the proposed algorithm to this fixed parameter, we performed the retrievals by adding a large error on this parameter (2000 m). Figure 5 shows these retrievals. Using the 3-channels approach, for cloud top altitude of 1 km, the algorithm do not converge. For low-level clouds with cloud top altitudes of 2 and 3 km, large absolute errors are observed (around -5 to -2 kg.m-2, respectively for moderate COTs, purple to blue lines). Above these altitudes, for cloud tops from 4 to 10 km, this parameter does not significantly modify the results, and absolute errors are in the same range as previously."

"In line 235 it is stated that "Consequently, less radiation is absorbed within the cloud with the SAS model used for the retrieval than with the AFGL tropical model used to simulate the test measurements." This seems to contradict the statement in line 215 that "different first guess profiles, both in idealized and realistic scenarios, give similar results." What is the real influence of the assumed profile?"

There is indeed confusion, we made a mistake here. Below the correct paragraph with additional information to discuss the influence of the assumed first guess profile:

(new file, page.10 lines.232-236) "Tests carried out with different first guess profiles, both in idealized and realistic scenarios, show that initial profile has an impact on incloud water vapor since we assume that RH=100% between cloud base and cloud top altitudes. So, starting iterations with the smooth SAS profile leads to an underestimation of the in-cloud water vapor absorption as the SAS profile is drier than the tropical profile used to generate the test data."

(new file, pages.11-12 lines.253-266) "The absolute error is positive, indicating that the retrieved  $IWV\_AC$  is overestimated due to less in-cloud water vapor absorption. This occurs because the water vapor profile within the cloud is not adjusted during the retrieval process, and the first-guess profile (AFGL SAS) is drier than the target AFGL tropical profile. Consequently, less radiation is absorbed within the cloud with the AFGL SAS model used for retrieval than with the AFGL tropical model used to simulate the test measurements. To compensate for this lower absorption and minimize the difference between the forward model simulations F(x) and the measurements y, the retrieved integrated water vapor above the cloud is overestimated. For optically thick clouds, the same behavior appears to a lesser extent, since less radiation interacts with the in-cloud water vapor.

Tests made using the tropical profile as the first guess in order to have the same incloud water vapor absorption between the simulated measurements and the retrievals

leads to smaller errors for thin clouds since the COT is well retrieved. Conversely, larger negative errors for low-level and optically thick clouds occur, as the algorithm underestimates the extinction profile. Consequently, there is more in-cloud absorption and the algorithm underestimates water vapor above clouds. Based on these results, the use of the SAS profile as a first guess appears to introduce a compensatory effect that partially mitigates the systematic underestimation of large COT."

---

## Author Response (AR2)

Dear reviewer, dear editor,

Thanks for your comments and your suggestions. In the revised manuscript all the suggested reformulations are accounted for, and your question has been addressed. We have listed your questions and comments below (in red), then explained how we have addressed each point in the revised manuscript and provided new sentences (in green).

**Reviewer #2's feedback #2 - accepted subject to minor revisions**

I am satisfied with most of the replies to my comments and commend the authors for the revisions made to the manuscript to address them. However, my comment on the impact of the assumption that cloud particles above 4km altitude are ice was not sufficiently addressed in my view. The additional text added to the manuscript is:

"The cloud model phase used during the retrieval process can also be improved, particularly the assumption of 4 km made for the top of the liquid phase, for tests on mixed-phase clouds, given that supercooled liquid clouds can be present well above 4 km altitude. A better definition of the different cloud phases would allow for a better representation of the cloud's optical properties."

I think this assumption can have a significant impact. At this stage I am not asking for thorough simulations to quantify this impact, but I suggest the authors give a rough estimate of the impact or at least estimate whether this will lead to over- or underestimations for super-cooled liquid tops (for example for liquid cloud tops at - 20 degrees Celcius).

➔ We agree that the assumption that all cloud particles above 4 km are ice can have a non-negligible impact, particularly when supercooled liquid layers are present at higher altitudes. Without simulations, the quantitative evaluation of this impact cannot be obtained. Nevertheless, in the shortwave infrared (SWIR) range, the cloud optical thickness (COT) and effective radius retrievals are highly sensitive to the assumed thermodynamic phase. As shown by King et al. (2013), the SWIR bands at 1.6 μm, 2.1 μm, and 3.7 μm used in MODIS retrievals exhibit different sensitivities to droplet size and phase, with the 2.1-μm band being more influenced by deeper cloud layers and 3.7-μm band by smaller droplets near the cloud top. When an ice phase is assumed, the simulated reflectances in these bands would likely be lower, leading to an underestimation of COT and biases in retrieved microphysical parameters.

Moreover, King et al. (2013) emphasize that *"ice cloud retrievals are particularly sensitive to particle shape (habit) assumptions"* and that a robust phase classification is a crucial step before the derivation of optical properties. In earlier versions of their algorithm, clouds with undetermined phase were *"subsequently processed as if they were liquid water"*, which resulted in systematic biases in retrieved COT and effective radius. This demonstrates that phase assumptions directly affect the accuracy of SWIR-based retrievals. In our case, applying a fixed 4-km threshold between liquid and ice phases could introduce similar biases. For supercooled liquid tops (e.g., around −20°C), we can expect the retrievals to underestimate the cloud optical thickness and potentially misattribute part of the SWIR absorption to water vapor above the cloud. We have added the following paragraphs to the conclusion/perspectives section:

- *"Indeed, assuming that all cloud particles above 4 km are considered as ice may introduce non-negligible uncertainties. In the SWIR spectral range, this assumption is likely to lead to lower simulated reflectances, resulting in an underestimate of cloud optical thickness and biases in the retrieved microphysical parameters (e.g., King et al., 2013). Consequently, a similar effect can be expected in our case. We have shown that COT retrieval affects the IWV_AC retrieval, and since the representation of the cloud thermodynamic phase directly impacts COT retrieval, it consequently also indirectly influences the IWV_AC retrieval."* (new file, page.21 lines.432-437)

- *"To determine the cloud top phase, we plan to use the 670 nm channel from the CLOUD imager since the ratio between the visible and the SWIR non-absorbing channel (1.04 µm) provides information on the cloud top phase (Riedi et al., 2010). In case of liquid cloud, the liquid cloud algorithm would be applied and in case of cloud top ice phase, we will use climatological profiles to better constraint the transition altitudes."* (new file, pages.21-22 lines.438-441)

Furthermore, in the added text, I would remove "for tests on mixed-phase clouds" as I think that is a confusing addition.

➔ Regarding the terms "liquid clouds" and "mixed-phase clouds", in the new version, we have opted for the terms "low- /mid-level clouds" instead of "liquid clouds" and "high-level clouds" instead of "mixed-phase clouds." (new file, pages.[8, 18, 19, 21] lines.[183-184, 365-366, "caption figure 11", 431] – we have replaced "for tests on mixed-phase clouds" by "for tests on high-level clouds")

**Editor's feedback**

Thank you for your revised submission. I have received one further review of it from one of the original referees, and from that and my own reading would appreciate some further revisions related to ice clouds. The reviewer's comments are:

"I am satisfied with most of the replies to my comments and commend the authors for the revisions made to the manuscript to address them. However, my comment on the impact of the assumption that cloud particles above 4km altitude are ice was not sufficiently addressed in my view. The additional text added to the manuscript is:

"The cloud model phase used during the retrieval process can also be improved, particularly the assumption of 4 km made for the top of the liquid phase, for tests on mixed-phase clouds, given that supercooled liquid clouds can be present well above 4 km altitude. A better definition of the different cloud phases would allow for a better representation of the cloud's optical properties."

I think this assumption can have a significant impact. At this stage I am not asking for thorough simulations to quantify this impact, but I suggest the authors give a rough estimate of the impact or at least estimate whether this will lead to over- or underestimations for super-cooled liquid tops (for example for liquid cloud tops at - 20 degrees Celcius). Furthermore, in the added text, I would remove "for tests on mixed-phase clouds" as I think that is a confusing addition."

I agree with this reviewer and would appreciate it if you could further revise the manuscript to better account for these comments. Furthermore, please note these comments from the file review team:

"1. You used scientific abbreviations in the "Short summary" text (LES) and are kindly asked to provide at least one written out version. This does not apply to chemical elements."
➔ In the new version of the manuscript, the acronyms in the abstract have been removed.

"2. The ROR database lists the institution of the corresponding author but with a different city than given in the manuscript. Please clarify whether the ROR "Laboratoire d'Optique Atmosphérique (Villeneuve-d'Ascq, France)" is still correct."
➔ The correct affiliation since 2018 is indeed the one indicated in the paper: *Univ. Lille, CNRS, UMR 8518 - LOA - Laboratoire d'Optique Atmosphérique, F-59000 Lille, France*

Added references:
-   King, M. D., Platnick, S., Menzel, W. P., Ackerman, S. A., and Hubanks, P. A.: Spatial and Temporal Distribution of Clouds Observed by MODIS Onboard the Terra and Aqua Satellites, IEEE Transactions on Geoscience and Remote Sensing, pp. 3826–3852, https://doi.org/10.1109/TGRS.2012.2227333, 2013. (new file, page.25 lines.544-546)
-   Riedi, J., Marchant, B., Platnick, S., Baum, B. A., Thieuleux, F., Oudard, C., Parol, F., Nicolas, J.-M., and Dubuisson, P.: Cloud thermodynamic phase inferred from merged POLDER and MODIS data, Atmospheric Chemistry and Physics, pp. 11 851–11 865, https://doi.org/10.5194/acp-10-11851-2010, 2010. (new file, page.26 lines.576-578)

---

## Author Response (AR3)

27 Oct 2025
**Editor decision: Publish as is**
by Andrew Sayer

Public justification (visible to the public if the article is accepted and published):
Dear authors,

Thank you for your further revision; I am happy to accept this version for publication in AMT.

Best wishes,

Andrew

Dear reviewers, dear editor, dear editorial team,

Many thanks for your feedback on our paper entitled *Water Vapor Content Retrieval Under Cloudy Sky Conditions from SWIR Satellite Measurements in the Context of C³IEL Space Mission Project*, and thanks for accepting to publish our paper in your journal.

To answer the last comment:

24 Oct 2025
**Uploaded files validated**
by Katja Gänger

Notification to the authors:
Affiliation 3: please add the city and the country.

We've added the missing elements in the affiliation #3 in the final version of the manuscript. Here is the full affiliation: *"Institut Universitaire de France (IUF), Paris, France"*